# Viruses are a dominant driver of protein adaptation in mammals

**David Enard\*, Le Cai, Carina Gwennap, Dmitri A Petrov**

Department of Biology, Stanford University, Stanford, United States

**Abstract** Viruses interact with hundreds to thousands of proteins in mammals, yet adaptation against viruses has only been studied in a few proteins specialized in antiviral defense. Whether adaptation to viruses typically involves only specialized antiviral proteins or affects a broad array of virus-interacting proteins is unknown. Here, we analyze adaptation in ~1300 virus-interacting proteins manually curated from a set of 9900 proteins conserved in all sequenced mammalian genomes. We show that viruses (i) use the more evolutionarily constrained proteins within the cellular functions they interact with and that (ii) despite this high constraint, virus-interacting proteins account for a high proportion of all protein adaptation in humans and other mammals. Adaptation is elevated in virus-interacting proteins across all functional categories, including both immune and non-immune functions. We conservatively estimate that viruses have driven close to 30% of all adaptive amino acid changes in the part of the human proteome conserved within mammals. Our results suggest that viruses are one of the most dominant drivers of evolutionary change across mammalian and human proteomes.

## Introduction

A number of proteins with a specialized role in antiviral defense have been shown to have exceptionally high rates of adaptation (*Cagliani et al., 2011*; *Cagliani et al., 2012*; Elde et al., 2009; *Fumagalli et al., 2010*; *Kerns et al., 2008*; *Liu et al., 2005*; *Sawyer et al., 2004*; *Sawyer et al., 2005*; *Sawyer et al., 2007*; *Sironi et al., 2012*; *Vasseur et al., 2011*). One example is protein kinase R (PKR), which recognizes viral double-stranded RNA upon infection, halts translation, and as a result blocks viral replication (*Elde et al., 2009*). PKR is one of the fastest adaptively evolving proteins in mammals. Specific amino acid changes in PKR have been shown to be associated with an arms race against viral decoys for the control of translation (*Elde et al., 2009*).

However, PKR and other fast-evolving antiviral defense proteins may not be representative of the hundreds or even thousands of other proteins that interact physically with viruses (virus-interacting proteins or VIPs in the rest of this manuscript). Most VIPs are not specialized in antiviral defense and do not have known roles in immunity. Many of these VIPs play instead key functions in basic cellular processes, some of which might be essential for viral replication.

In principle some VIPs without specific antiretroviral functions might nonetheless evolve to limit viral replication or alleviate deleterious effects of viruses despite the need to balance this evolutionary response with the maintenance of the key cellular functions they play. There are reasons to believe that such an evolutionary response to viruses might be limited, however. First, most VIPs evolve unusually slowly rather than unusually fast both in animals (*Davis et al., 2015*; *Jäger et al., 2011*) and in plants (*Mukhtar et al., 2011*; *Weßling et al., 2014*). Second, VIPs tend to interact with proteins that are functionally important hubs in the protein-protein interaction network of the host possibly limiting their ability to adapt (*Dyer et al., 2008*; *Halehalli and Nagarajaram, 2015*). Finally, very few cases of adaptation to viruses are known outside of fast evolving, specialized antiviral proteins (*Demogines et al., 2012*; *Meyerson et al., 2014*; *Meyerson and Sawyer, 2011*;

**\*For correspondence:** denard@stanford.edu

**Competing interests:** The authors declare that no competing interests exist.

**eLife digest** When an environmental change occurs, species are able to adapt in response due to mutations in their DNA. Although these mutations occur randomly, by chance some of them make the organism better suited to their new environment. These are known as adaptive mutations.

In the past ten years, evolutionary biologists have discovered a large number of adaptive mutations in a wide variety of locations in the genome – the complete set of DNA – of humans and other mammals. The fact that adaptive mutations are so pervasive is puzzling. What kind of environmental pressure could possibly drive so much adaptation in so many parts of the genome?

Viruses are ideal suspects since they are always present, ever-changing and interact with many different locations of the genome. However, only a few mammalian genes had been studied to see whether they adapt to the presence of viruses. By studying thousands of proteins whose genetic sequence is conserved in all mammalian species, Enard et al. now suggest that viruses explain a substantial part of the total adaptation observed in the genomes of humans and other mammals. For instance, as much as one third of the adaptive mutations that affect human proteins seem to have occurred in response to viruses.

So far, Enard et al. have only studied old adaptations that occurred millions of years ago in humans and other mammals. Further studies will investigate how much of the recent adaptation in the human genome can also be explained by the arms race against viruses.

*Meyerson et al., 2015*; *Ng et al., 2015*; *Ortiz et al., 2009*; *Schaller et al., 2011*). Transferrin receptor or TFRC is the most notable exception, and serves as a striking example of a non-immune, housekeeping protein used by viruses (*Demogines et al., 2013*; *Kaelber et al., 2012*). TFRC is responsible for iron uptake in many different cell types and is used as a cell surface receptor by diverse viruses in rodents and carnivores. TFRC has repeatedly evaded binding by viruses through recurrent adaptive amino acid changes. As such, TFRC is the only clear-cut example of a host protein not involved in antiviral response that is known to adapt in response to viruses.

Here we analyze patterns of evolutionary constraint and adaptation in a high quality set of ~1300 VIPs that we manually curated from virology literature. These 1300 VIPs come from a set of ~10000 proteins conserved across 24 well-sequenced mammalian genomes (Materials and methods). As expected, the vast majority of these VIPs (~80%) have no known antiviral or any other more broadly defined immune activity. We confirm that VIPs do tend to evolve slowly and demonstrate that this is because VIPs experience much stronger evolutionary constraint than other proteins within the same functional categories. However, despite this greater evolutionary constraint, VIPs display higher rates of adaptation compared to other proteins. This excess of adaptation is visible in VIPs across biological functions, on multiple time scales, in multiple taxa, and across multiple studied viruses. Finally, we showcase the power of our global scan for adaptation in VIPs by studying the case of aminopeptidase N, a well-known multifunctional enzyme (*Mina-Osorio, 2008*) used by coronaviruses as a receptor (*Delmas et al., 1992*; *Yeager et al., 1992*). Using our approach we reach an amino-acid level understanding of parallel adaptive evolution in aminopeptidase N in response to coronaviruses in a wide range of mammals.

## Results

### Identification of VIPs

We curated a set of 1256 VIPs from the low-throughput virology literature (Materials and methods and *Supplementary file 1A*). VIPs were defined as proteins that interact physically with viral proteins, viral RNA, and/or viral DNA (*Supplementary file 1A*). We excluded interactions identified by high-throughput experiments because we were concerned about a high rate of false positives (*Mellacheruvu et al., 2013*). The 1256 VIPs were annotated from an initial set of 9861 proteins with clear orthologs in all 24 analyzed mammalian high quality genomes (*Figure 1*, *Supplementary file 1B* and Materials and methods) (*Enard et al., 2016*).

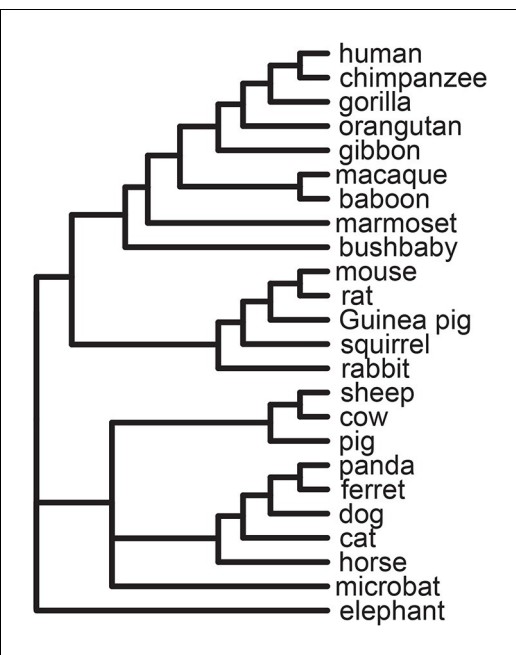

**Figure 1.** Tree of 24 mammals used in the analysis.

Most of the VIPs (95%) correspond to an interaction between a human protein and a virus infecting humans (*Supplementary file 1A*). Human Immunodeficiency Virus type 1 (HIV-1) is the best-represented virus with 240 VIPs, with nine other viruses (HPV, HCV, EBV, HBV, HSV, Influenza Virus, ADV, HTLV and KSHV) having at least 50 VIPs (*Supplementary file 1A*).

This dataset represents the largest, most up-to-date set of VIPs backed by individual low-throughput publications. Nonetheless, given that many VIPs were discovered only recently, with half of all publications reporting VIPs published in the past 7 years (*Figure 2*), it is likely that many additional VIPs remain to be discovered.

## Basic description of the identified VIPs

The identified 1256 VIPs are involved in diverse cellular and supracellular processes with 162 overlapping GO cellular and supracellular processes having more than 50 VIPs (Gene Ontology (GO) classes (October 2013 version) (*Ashburner et al., 2000*; *The Gene Ontology Consortium, 2015*); *Supplementary file 1C*). These cellular processes include transcription (354 VIPs), post-translational protein modification (224 VIPs), signal transduction (396 VIPs), apoptosis (185 VIPs), and transport (264 VIPs). The supracellular processes notably include defense response (103 VIPs) and developmental processes (327 VIPs). Only 57 VIPs or 5% of VIPs have known antiviral activity (*Supplementary file 1D*). These 57 antiviral VIPs are part of a larger group of 241 VIPs (20% of VIPs) with known immune functions, defined here as any activity that modulates the immune response or involved in the development of the immune response (Materials and methods and *Supplementary file 1D*). Most - more than 80% - of the VIPs have no known immune activity.

## Overview of the evolutionary analysis of VIPs versus non-VIPs

We analyze both purifying selection and positive selection in VIPs versus non-VIPs at two distinct evolutionary time scales: (i) in the great apes in general and in the human branch specifically and (ii) across the entire mammalian phylogeny. We use the ratio of nonsynonymous to synonymous polymorphisms (abbreviated as pN/pS) within humans and great apes as a measure of purifying selection. We use McDonald-Kreitman (MK) and the branch-site tests of positive selection using the BS-REL (*Kosakovsky Pond et al., 2011*) and BUSTED (*Murrell et al., 2015*) tests from the HYPHY package (*Pond et al., 2005*) to assess the prevalence of positive selection in VIPs compared to non-VIPs in the human lineage and in mammals in general (Material and methods).

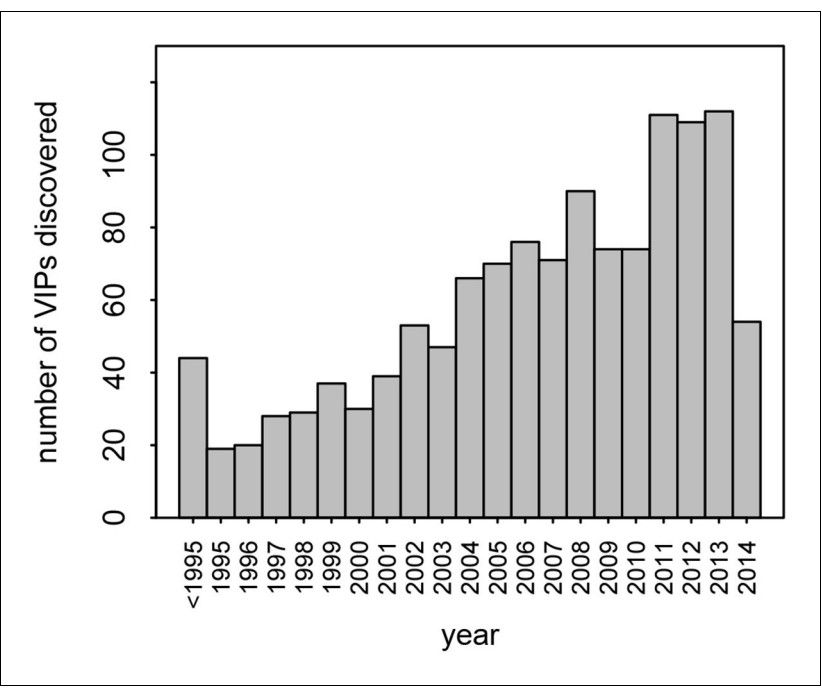

**Figure 2.** Number of VIPs discovered per year until 2014.

## Patterns of purifying selection in VIPs

We confirm that VIPs tend to evolve slowly (*Jäger et al., 2011*; *Davis et al., 2015*). On average, the VIPs have ~15% lower mammal-wide dN/dS ratio compared to non-VIPs (0.124 versus 0.145, 95% CI [0.136,0.148]; Materials and methods). The difference in dN/dS is highly significant (permutation test $P=0$ after $10^9$ iterations; *Supplementary file 1B*). In order to disentangle whether this slower evolution of VIPs is due to stronger purifying selection or to a lower rate of adaptation, we first assess the strength of purifying selection in the VIPs using the pN/pS ratio.

Genome-wide polymorphism data required to measure pN/pS are available in humans (*Abecasis et al., 2012*) (1000 Genomes Project) (*Supplementary file 1E*), and other great apes: chimpanzee, gorilla, and orangutans (*Prado-Martinez et al., 2013*) (Great Apes Genome Project) (*Supplementary file 1F*). The 1000 Genomes Project and the Great Apes Genome Project are complementary for this analysis. On the one hand, the 1000 Genomes Project provides high quality variants with frequencies estimated from a large number of individuals. On the other hand while the Great Ape Genome project includes fewer individuals and provides coarser frequency data, it provides substantially higher pN and pS counts than the 1000 Genomes data because non-human great apes tend to be more polymorphic overall (*Prado-Martinez et al., 2013*).

In the human African populations from the 1000 Genomes project (Materials and methods), the average pN/pS is 21% lower in VIPs compared to non-VIPs (0.759 versus 0.966, 95% CI [0.92,1.01], simple permutation test $P=0$ after $10^9$ iterations). VIPs also show an excess of low frequency ($\leq$10%) deleterious non-synonymous variants compared to non-VIPs (*Figure 3—figure supplement 1*; simple permutation test $P=0$ after $10^9$ iterations). In great apes, the average pN/pS ratio is 25% lower in VIPs compared to non-VIPs (0.526 versus 0.697, 95% CI [0.66,0.72], simple permutation test $P=0$ after $10^9$ iterations; *Figure 3A*). Finally, stronger purifying selection acting on VIPs is widespread and is not limited to VIPs interacting with any one particular virus (*Figure 3B*).

VIPs and non-VIPs have slightly different coding sequence GC content (0.516 versus 0.523 on average, $P=6\times10^{-4}$), coding sequence lengths (668 versus 606 amino acids on average, $P=0$) and recombination rates (*Kong et al., 2010*) (1.145 cM/Mb versus 1.175 cM/Mb on average, $P=0.21$). To ensure that the difference in pN/pS between VIPs and non-VIPs is robust to these differences, we compare VIPs with non-VIPs with similar values for each potential confounding factor using permutations with a target average (Materials and methods). The difference in pN/pS in great apes between

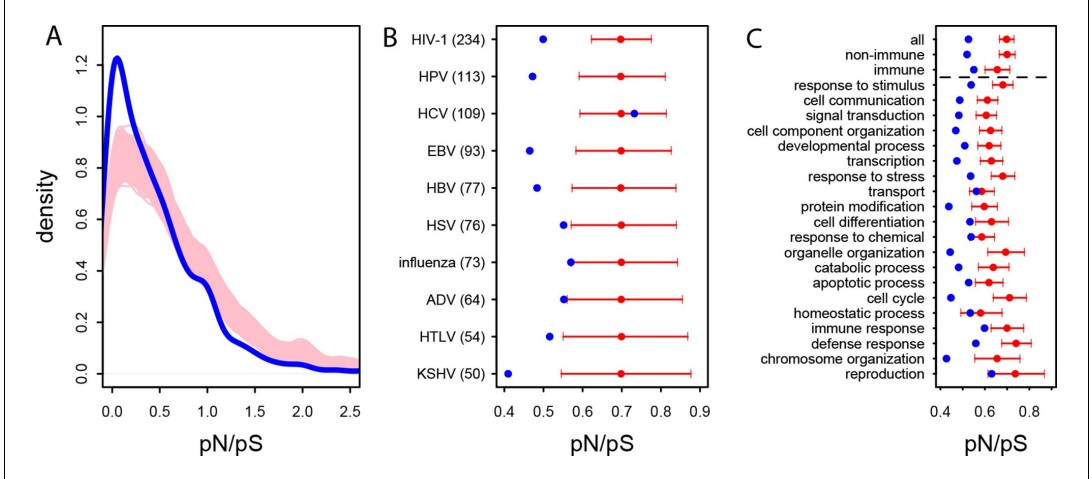

**Figure 3.** Patterns of purifying selection in VIPs. (**A**) Distribution of pN/pS in VIPs (blue) and non-VIPs (pink). The blue curve is the density curve of pN/(pS+1) for 1256 VIPs. We use pN/(pS+1) instead of pN/pS to account for those coding sequences where pS=0. pN and pS are measured using great ape genomes from the Great Ape Genome Project (Materials and methods). The pink area represents the superimposition of the density curves for each of 5000 sets of randomly sampled non-VIPs. (**B**) Average pN/pS in VIPs (blue dot) versus average pN/pS in non-VIPs (red dot and red 95% confidence interval) within ten viruses with more than 50 VIPs The number between parentheses is the number of VIPs for each virus. KSHV: Kaposi's Sarcoma Herpesvirus. HIV-1: Human Immunodeficiency Virus type 1. HBV: Hepatitis B Virus. ADV: Adenovirus. HPV: Human Papillomavirus. HSV: Herpes Simplex Virus. EBV: Epstein-Barr Virus. Influenza: Influenza Virus. HTLV: Human T-lymphotropic Virus. HCV: Hepatitis C virus. (**C**) Same as B), but for the 20 most high level GO processes with the highest number of VIPs. The full GO process name for "protein modification" as written in the figure is "post-translational protein modification".

The following figure supplement is available for figure 3:

**Figure supplement 1.** Site Frequency Spectrum of non-synonymous variants in VIPs and non-VIPs in African populations Red: VIPs.

VIPs and non-VIPs persists when comparing VIPs and non-VIPs with similar GC content (0.526 versus 0.655, $P$=0 after $10^9$ iterations), similar coding sequence length (0.526 versus 0.654, $P$=0), or similar recombination (0.526 versus 0.702, $P$=0). The difference in pN/pS between VIPs and non-VIPs is therefore a genuine difference in the strength of purifying selection and not due to confounding factors biasing the pN/pS ratio.

VIPs have been shown before to be broadly expressed genes and to serve as hubs in the human protein-protein interactions network (*Dyer et al., 2008*, *Halehalli and Nagarajaram, 2015*). These differences in gene expression and the number of protein-protein interactions may explain the stronger purifying selection experienced by VIPs. We confirm that VIPs are indeed expressed in more tissues than non-VIPs both at the RNA level (*GTEx Consortium, 2015*) (GTEx V4 RNA-seq expression RPKM≥10 in 25.5 tissues on average in VIPs versus 11.9 tissues in non-VIPs, simple permutation test $P$=0) and at the protein level (*Kim et al., 2014*) (Human Proteome Map spectral count≥5 in 15.1 tissues on average for VIPs versus 6.1 for non-VIPs, simple permutation test $P$=0). VIPs also have many more protein-protein interaction partners than non-VIPs based on a dataset of human protein-protein interactions curated by (*Luisi et al., 2015*) from the Biogrid database (*Stark et al., 2011*) (18.4 on average versus 3.2, simple permutation test $P$=0).

The magnitude of the difference in pN/pS between VIPs and non-VIPs expressed in a similar number of tissues at the RNA level (GTEx) (0.526 versus 0.647, $P$=0) or in a similar number of tissues at the protein level (Human protein Map) (0.526 versus 0.662, $P$=0) remains largely unchanged. In contrast, the difference in pN/pS is strongly affected when comparing VIPs and non-VIPs with a similar number of protein-protein interactions. Indeed, non-VIPs with the same number of interacting partners as VIPs have a pN/pS ratio of 0.605 versus 0.697 for all non-VIPs, and the difference in the pN/pS ratios between VIPs and non-VIPs is reduced from 25% to 13%. These results show that VIPs do experience stronger purifying selection than non-VIPs, and that the difference in purifying selection is driven at least partly by the fact that VIPs tend to be hubs with many interacting partners in the human protein-protein interactions network.

The higher level of purifying selection in VIPs might be due to the fact that VIPs participate in the more constrained host functions, or, alternatively, because within each specific host function, viruses tend to interact with the more constrained proteins. In order to test these two non-mutually exclusive scenarios we generated $10^4$ control sets of non-VIPs chosen to be in the same 162 Gene Ontology processes as VIPs (GO processes with more than 50 VIPs; *Supplementary file 1C* and Materials and methods). In great apes, GO-matched non-VIPs still have a much higher pN/pS ratio compared to VIPs, suggesting that VIPs tend to be more conserved than non-VIPs from the same GO category. On average, pN/pS in the GO-matched non-VIPs is 0.647 (95% CI [0.621,0.674]). This is only slightly lower than the average ratio in non-VIPs in general (pN/pS=0.697, $P=2\text{x}10^{-3}$), but much higher than the average ratio in VIPs (0.526, permutation test $P=0$ after $10^4$ iterations). Moreover, the stronger purifying selection acting on VIPs is apparent within most functions. *Figure 3C* shows stronger purifying selection in the 20 high level GO categories with the most VIPs. In all the 20 GO categories pN/pS is lower in VIPs than in non-VIPs, and the difference is significant for 17 of these categories (*Supplementary file 1C*). This shows that within a wide range of host functions, viruses tend to interact with the most conserved proteins.

Interestingly, even immune VIPs (*Supplementary file 1D*) have a significantly reduced pN/pS ratio compared to immune non-VIPs (*Figure 3C*), which suggests that immune proteins in direct physical contact with viruses are more constrained. The reduction in pN/pS in non-immune VIPs is very similar to the reduction observed in the entire set of VIPs (*Figure 3C*). The table at *Supplementary file 1C* further shows stronger purifying selection in 124 of the 162 GO categories (77%) with more than 50 VIPs.

## Frequent adaptation in VIPs in the human lineage
### The classic MK test in the human lineage

We estimate the proportion of adaptive non-synonymous substitutions (noted $\alpha$) in VIPs and non-VIPs in the human lineage by using the classic McDonald-Kreitman test (MK test) (*McDonald and Kreitman, 1991*) (Materials and methods). We use the 1000 Genomes Project polymorphism data from African populations (Materials and methods and *Supplementary file 1E*) and divergence between humans and chimpanzees. We first attempt to limit the effect of deleterious variants by excluding all variants with a derived allele frequency lower than 10% (Materials and methods) (*Keightley and Eyre-Walker, 2007*, *Charlesworth and Eyre-Walker, 2008a*, *Eyre-Walker and Keightley, 2009*, *Messer and Petrov, 2013*). We find that $\alpha$ is strongly elevated in VIPs compared to non-VIPs ($\alpha$=0.19 in VIPs versus $-0.02$ in non-VIPs, permutation test $P=2.\text{x}10^{-5}$).

Note that the classic MK test is known to underestimate the true $\alpha$ in the presence of slightly deleterious polymorphisms (*Charlesworth and Eyre-Walker, 2008b*). Given that VIPs tend to have more non-synonymous deleterious low frequency variants than non-VIPs (*Figure 3—figure supplement 1*) this downward bias should be stronger in the VIPs, making this comparison conservative and indicating that VIPs likely have a substantial excess of adaptation compared to non-VIPs.

The difference in $\alpha$ is robust to recombination ($\alpha$ =$-0.025$ in non-VIPs with similar recombination to VIPs versus $-0.02$ without control). It is also robust to coding sequence GC content ($\alpha$=$-0.019$ with versus $-0.02$ without control), coding sequence length ($\alpha$=$-0.023$ with versus $-0.02$ without control). The difference is also robust to variation in levels of expression at the RNA level measured as the number of tissues with GTEx V4 RNA-seq expression RPKM$\geq$10 ($\alpha$=0.001 with versus 0.02 without control) and as the average expression across all GTEx V4 tissues ($\alpha$=$-0.018$ with versus 0.02 without control), as well as at the protein level measured as the number of Human Proteome Map tissues with spectral count>=5 ($\alpha$=$-0.024$ with versus 0.02 without control) or the average expression across all the Human Proteome Map tissues ($\alpha$=$-0.035$ with versus 0.02 without control). The difference in $\alpha$ is also not affected by the number of protein-protein interactions ($\alpha$=0.025 with versus $-0.02$ without control). The difference in $\alpha$ is not affected either by purifying selection, as shown by the fact that using great apes pN/pS or human pN/pS as a control has no effect ($\alpha$=0.001 with versus 0.02 without control in both cases). Finally, we match VIPs and non-VIPs with similar GO categories (Materials and methods, paragraph titled 'Gene Ontology-matching control samples'). The higher rate of adaptation in VIPs is not explained by higher rates of adaptation in the host GO processes where VIPs are well represented ($\alpha$=0.003 with versus $-0.02$ without control). For all the controls, the difference in $\alpha$ between VIPs and non-VIPs remains highly significant (permutation test

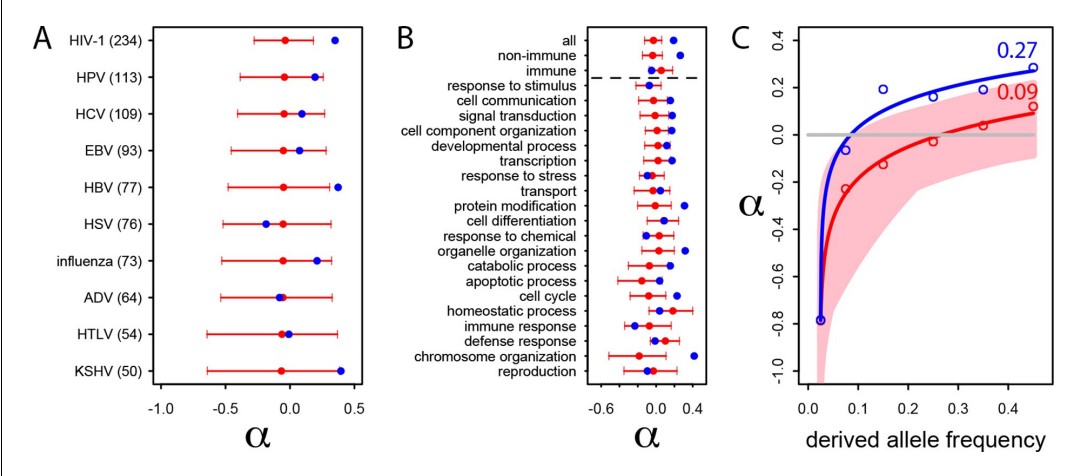

**Figure 4.** Patterns of human adaptation in VIPs. (A) Classic MK test (Materials and methods) for VIPs (blue dot) and non-VIPs (red dot and 95% confidence interval) for the ten viruses with 50 or more VIPs. (B) Same as A) but for the 20 top high level GO processes with the most VIPs below the dotted black line. Above the dotted black line: the classic MK test for all VIPs, for non-immune VIPs and for immune VIPs (*Supplementary file 1D*). (C) Asymptotic MK test (Materials and methods) for the proportion of adaptive amino acid substitutions ($\alpha$) in VIPs (blue dots and curve) and non-VIPs (red dots and curve). Pink area: superposition of fitted logarithmic curves (Materials and methods) for 5000 random sets of 1256 non-VIPs (as many as VIPs) where the estimated $\alpha$ falls within $\alpha$'s 95% confidence interval.

$P<10^{-3}$ in all cases). Together these results show that the excess of adaptation in VIPs is robust to many different host factors.

We further investigate the excess of adaptation for the specific VIPs of ten human viruses and in the 20 high level GO categories with the most VIPs (*Figure 4A and B*). Although the small number of proteins interacting with individual viruses precludes precise estimates of $\alpha$ (see the large confidence intervals on *Figure 4A*), the VIPs show nominally higher values of $\alpha$ for eight out of 10 viruses, with HIV-1 and Hepatitis B Virus (HBV) displaying statistically significant increases in adaptation. Likewise, VIPs in most GO categories show higher rates of adaptation (14 out of 20) with 9 of 14 showing statistically significant increases (*Figure 4B*).

Finally and importantly, the 80% of VIPs with no known antiviral or broader immune function (*Supplementary file 1D*) have a strongly increased rate of adaptation according to the classic MK test ($\alpha=0.26$ in VIPs versus 0.02 in non-VIPs, permutation test $P=3\times10^{-7}$; *Figure 2B*). Intriguingly, unlike for non-immune VIPs or all VIPs considered together (top of *Figure 4B*), immune VIPs, including antiviral VIPs (*Supplementary file 1D*), do not show any increase of adaptation compared to immune non-VIPs. The lack of a signal is unlikely to be due to reduced statistical power of the comparison in a smaller set of immune proteins, given that 1000 random samples of non-immune VIPs with the same size as the immune VIPs sample (241) always exhibited a significantly ($p<0.05$) increased rate of adaptation compared to non-immune non-VIPs.

## The asymptotic MK test

The classic MK test is known to be biased downward by the presence of slightly deleterious non-synonymous variants (*Charlesworth and Eyre-Walker, 2008b*) and this bias is difficult to eliminate fully even by excluding low frequency variants (*Messer and Petrov, 2013*). Messer and Petrov suggested an asymptotic modification of the MK test which provides less biased estimates of $\alpha$ in the presence of slightly deleterious variants (*Messer and Petrov, 2013*). The Messer-Petrov approach estimates $\alpha$ for each frequency category separately and then uses the functional dependence of $\alpha$ on allele frequency to extrapolate $\alpha$ at fixation. This approach thus requires well-resolved frequency data necessitating the use of the 1000 genomes data and also lacks power to estimate $\alpha$ for small subsets of genes. We thus use this approach only to better quantify the true $\alpha$ in the complete sets of VIPs and non-VIPs. To further validate the asymptotic MK test we carry out extensive population simulations

using SLiM (*Messer, 2013*) and show that this test is robust to demographic events such as bottlenecks or population expansions (Materials and methods and *Supplementary file 1G*).

The asymptotic MK test estimate of $\alpha$ in VIPs is 27% (512 out of 1897 amino acid changes) compared to ~9% (1293 out of 14,370 amino acid changes) in non-VIPs (*Figure 4C*). Thus, although VIPs represent only 13% of the orthologs in our dataset and only 11% of all amino acid substitutions, we estimate that in human evolution they account for almost 30% of all adaptive amino-acid changes. Note that both VIPs and non-VIPs in our dataset are limited to the proteins conserved across all mammals.

## Frequent adaptation in VIPs across mammals

The increased rate of adaptation in VIPs in the human lineage strongly suggests that VIPs in our dataset, 95% of which interact with modern viruses affecting humans (*Supplementary file 1A*), were also VIPs during the last 7 million years of human evolution since the split with chimpanzees. It is also plausible that a substantial proportion of the VIPs we study are also VIPs in multiple mammalian lineages. Indeed, viruses infecting humans (including the ten viruses with the most VIPs) are known to have close viral relatives in many other mammals, with the exception of Hepatitis C Virus (HCV) for which only distant relatives are known and primarily in bats (*Quan et al., 2013*). There is also growing evidence that distantly related viruses tend to interact with overlapping sets of host proteins (*Jäger et al., 2011*, *Davis et al., 2015*). We thus hypothesize that VIPs, while identified primarily in humans, may have also experienced frequent adaptation in mammals in general, with the possible exception of the VIPs interacting with HCV.

To test this hypothesis we use the Branch-Site Random Effect Likelihood test (BS-REL test) (*Kosakovsky Pond et al., 2011*) and the BUSTED test (*Murrell et al., 2015*) both available in the HYPHY package (*Pond et al., 2005*) in order to detect episodes of adaptive evolution in each of the 44 branches of the mammalian tree used for the analysis (Materials and methods). For a specific coding sequence, the BS-REL and BUSTED tests estimate the proportion of codons where the rate of non-synonymous substitutions is higher than the rate of synonymous substitutions (dN/dS>1), which is a hallmark of adaptive evolution. The BS-REL test estimates proportions of selected codons specifically for each branch, whereas BUSTED estimates an overall proportion of selected codons across the entire tree. Both tests then compare two competing models of evolution, one with adaptive substitutions and one without adaptive substitutions, and decide whether the model with adaptation is a significantly better fit to the data.

The BUSTED *P*-value is a good measure of whether a specific protein experienced adaptation in the history of mammalian evolution. In addition to presence/absence of adaptation, we assess the amount of adaptation experienced by a particular protein by estimating the average proportion of selected codons from the BS-REL test along all mammalian branches.

We compare the proportion of selected codons detected by the BS-REL test between VIPs and non-VIPs. The statistical power of BUSTED and the BS-REL test has been shown to depend strongly on the amount of constraint in a coding sequence, with higher constraint/purifying selection decreasing the ability to detect adaptation (*Kosakovsky Pond et al., 2011*). We confirm this in our dataset by observing a strong positive correlation between the pN/pS ratio in great apes and the proportion of selected codons across mammals estimated by the BS-REL test (Spearman's rank correlation $\rho$ =0.34, p<2x10$^{-16}$, n=9861). We therefore use a permutation test with a target average (Materials and methods) that matches VIPs and non-VIPs with similar pN/pS ratios in order to compare VIPs and non-VIPs that experience similar levels of purifying selection and providing us with similar power to detect adaptation (Materials and methods and *Figure 5—figure supplements 1* and *2*).

The permutation test shows that adaptation has been much more common in VIPs than in non-VIPs across mammals (*Figure 5*). We estimate that all VIPs have experienced twice as many adaptive amino acid changes on average compared to non-VIPs (*Figure 5A*, permutation test *P*=0 after 10$^9$ iterations). We further use an increasingly strict level of evidence for the presence of adaptation, by including only proteins with increasingly low BUSTED *P*-values; that is, increasingly high probability that adaptation occurred somewhere on the tree (*Figure 5A*). *Figure 5A* shows that VIPs with the strongest evidence of adaptation (BUSTED *P*-values lower than 10$^{-5}$) have a six-fold excess of strong signals of adaptation (permutation test *P*=0 after 10$^9$ permutations). In *Figure 5—figure supplement 3* we further show that this excess of adaptation in VIPs is due to i) more VIPs with signals of adaptation than non-VIPs, ii) more branches of the tree per VIP showing adaptation, and iii) a greater

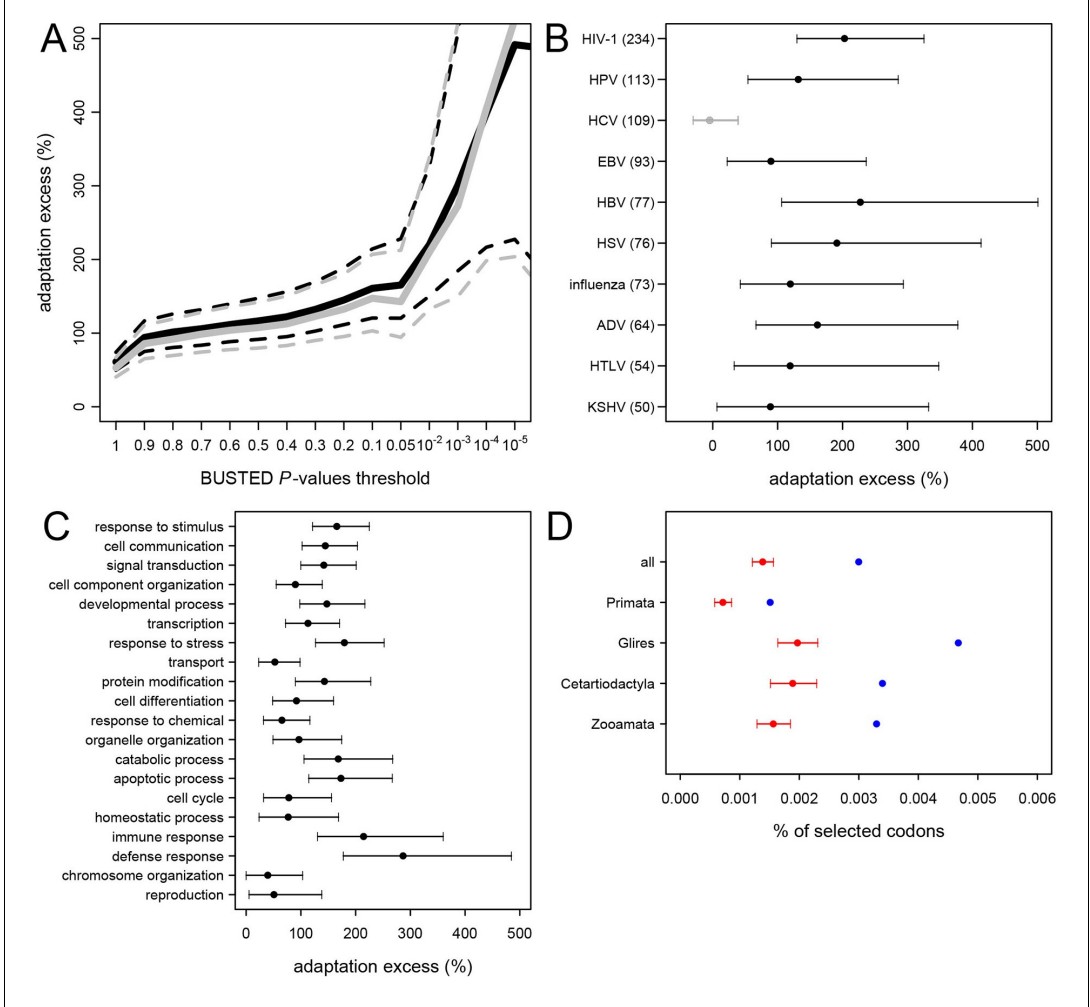

**Figure 5.** Excess of adaptation across mammals in VIPs The excess of adaptation is measured as the extra percentage of adaptation in VIPs compared to non-VIPs. For example, if VIPs have 1.5 times or 50% more adaptation, then the adaptation excess is 50%. (**A**) Thick black curve: average excess of adaptation in all VIPs. Dotted black curves: 95% confidence interval for the excess of adaptation in all VIPs. Thick grey curve: excess of adaptation in non-immune VIPs. Dotted grey curves: 95% confidence interval for the excess of adaptation in non-immune VIPs. (**B**) Virus-by-virus excess of adaptation in VIPs. Black dot is the average excess and the represented interval is the 95% confidence interval. Excess is shown for BUSTED p≤0.5. (**C**) Excess of adaptation within the top 20 high-level GO processes with the most VIPs. Excess is shown for BUSTED p≤0.5. (**D**) Proportions of selected codons in VIPs (blue dot) and non-VIPs (red dot and 95% confidence interval) in the mammalian clades represented by more than one species in the tree. All: entire tree. Primata: primates. Glires: rodents and rabbit. Cetartyodactyla: sheep, cow, pig. Zooamata: carnivores and horse. Excess is shown for BUSTED p≤0.5.

The following figure supplements are available for figure 5:

**Figure supplement 1.** How to compare VIPs and non-VIPs across mammals Red: part of dN/dS explained by adaptive evolution.

**Figure supplement 2.** Scheme for the permutation test with a target average using the example of purifying selection.

**Figure supplement 3.** Contributions of the number of genes, number of branches and proportion of selected codons to the excess of adaptation in VIPs.

proportion of codons evolving adaptively per branch. In line with the MK test, we find that the excess of adaptation in mammals is robust to the potential confounding factors of expression at the RNA and protein levels, and to the number of host protein-protein interactions (*Supplementary file 1H*). Indeed, adaptation in mammals remains at least twice more frequent in VIPs compared to non-

VIPs expressed at the RNA level in many tissues (permutation test p<10-5for VIPs and non-VIPs expressed in at least 10, 20, 30 or 40 GTEx tissues), VIPs and non-VIPs expressed at the protein level in many tissues (permutation test p<for VIPs and non-VIPs expressed in at least 10 or 20 Human Proteome Map tissues), and VIPs and non-VIPs with a high number of protein-interacting partners (permutation test $P<10^{-5}$for VIPs and non-VIPs with at least 5 or 10 protein-interacting partners).

## Adaptation to specific viruses

We further quantify the excess of adaptation specifically for each of the ten viruses with more than 50 VIPs (HIV-1, HPV, HCV, EBV, HBV, HSV, Influenza Virus, ADV, HTLV and KSHV) (*Figure 5B*). Nine out of the ten viruses show a strong excess of adaptation in their respective VIPs. HIV-1 and HBV VIPs have the strongest excess, with three times as many selected codons as non-VIPs.

As mentioned above, HCV stands out among the ten viruses with the largest number of VIPs in humans in that it has no known close viral relatives despite extensive screening of diverse mammalian species (*Quan et al., 2013*). If this reflects a true lack of close viral relatives of HCV, then we predict a limited excess of adaptation in HCV VIPs. In line with this prediction, the 109 VIPs of HCV are the only ones where we do not detect any excess of adaptation along the mammalian tree (*Figure 5B*) despite being one of the largest groups of VIPs. VIPs that interact with all other viruses all show substantial elevation of adaptation (*Figure 5B*).

## GO-analysis of adaptation in mammals

VIPs are represented in a wide range of GO functions with 162 GO categories having more than 50 VIPs (*Supplementary file 1C*). Of these 162 GO categories, 118 (73%) have a 50% or greater excess of adaptation in VIPs (*Supplementary file 1C*, permutation test p<0.05 in all cases). The excess of adaptation in VIPs is therefore widespread across host functions. GO processes with a strong excess of adaptation include cellular processes such as transcription, signal transduction, apoptosis, or post-translational protein modification, but also supracellular processes related to development (*Figure 5C* and *Supplementary file 5*). As expected, immune VIPs have a very strong excess of adaptation compared to immune non-VIPs (*Figure 5C*). Importantly, VIPs with no known immune function (*Supplementary file 1D*) show a very similar excess of adaptation compared to all VIPs (*Figure 5A*, black versus grey lines; permutation test $P=0$ after $10^9$ iterations). Overall these results suggest that the arms race with viruses has strongly increased the rate of adaptation in a wide range of VIPs.

## Distribution of adaptation in VIPs across mammalian clades

Since 95% of the VIPs were discovered for viruses infecting humans, it is possible that the observed excess of adaptation in VIPs in mammals is due to higher rates of adaptation exclusively in the primate branches of the mammalian tree (*Figure 1*). However, all mammalian clades in the tree show a similar excess of adaptation in VIPs (*Figure 5D*). Primates stand out due to their low overall proportions of positively selected codons compared to the other mammalian clades in the tree (*Figure 5D*). This is most likely due to a lower statistical power of the BS-REL test in the short primate branches (*Kosakovsky Pond et al., 2011*). In line with this, VIPs with strong signals of adaptation show such signals in all the mammalian clades represented (*Figure 6*). This includes well-known antiviral VIPs (*Figure 6A*), antiviral VIPs where adaptation was previously unknown across mammals (*Figure 6B*), and non-antiviral VIPs with diverse, well-studied functions in the mammalian hosts (*Figure 6C*). This phylogenetically widespread excess of recurrent adaptation suggests that many of the VIPs annotated in humans were also VIPs for a substantial evolutionary time in a wide range of mammals.

## From global patterns of adaptation to understanding specific instances of adaptation to viruses: the case of coronaviruses and aminopeptidase N

We have shown that rates of adaptation are globally elevated in VIPs in humans and mammals in general, suggesting the existence of tens of thousands of isolated events of adaptations to a diverse range of viruses. Here, we test if our global approach has enough power to isolate new specific cases of adaptation to viruses by looking for instances where viruses are the plausible cause of

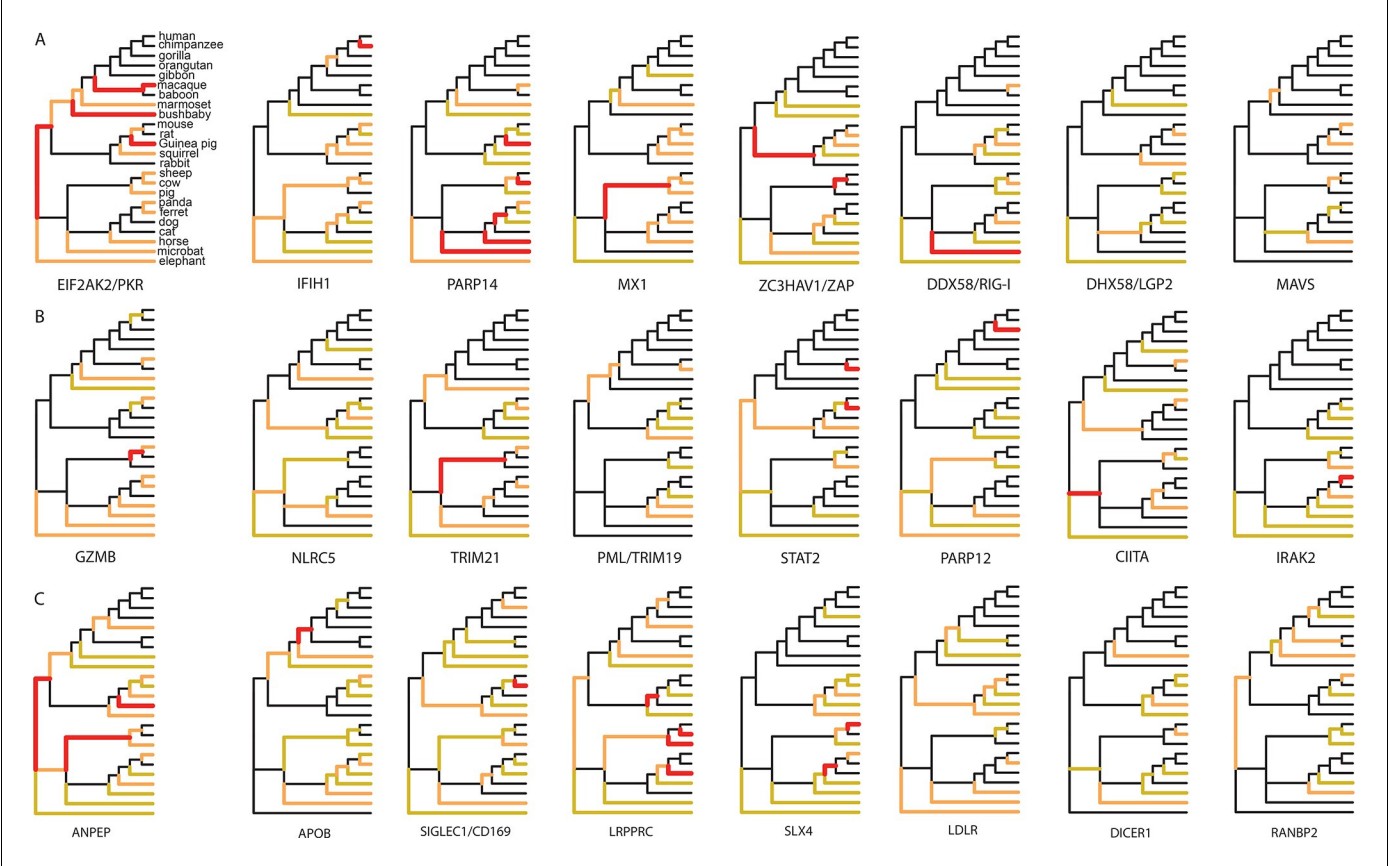

**Figure 6.** Examples of mammalian orthologs with adaptation spread across clades. (A) Signals of adaptation in eight antiviral proteins with well-known adaptation across mammals. Red: BS-REL p≤0.001. Orange: BS-REL p≤0.05. Yellow: BS-REL p≤0.1. (B) Top eight antiviral proteins with the highest number of branches under selection, and no previously know adaption spread across mammals. Note that adaptation was previously found for TRIM21 in primates but no other mammalian clade (*Malfavon-Borja et al., 2013*). (C) Top eight non-antiviral proteins with well-known functions and the highest number of branches under selection across mammals. Proteins are ordered according to the number of branches with signals of adaptation.

adaptation in a VIP with no known antiviral activity. This is particularly relevant because, to our knowledge, the transferrin receptor (TFRC) is one of the only well understood case of a non-antiviral protein adapting in response to viruses (*Demogines et al., 2013*).

To identify a non-antiviral VIP for in-depth investigation we first excluded all VIPs with a well-known antiviral activity (*Supplementary file 6*; here as in the rest of the manuscript antiviral means a protein activity that restricts viral infection) and then selected all remaining VIPs with strong overall evidence of adaptation (*Supplementary file 1I*) and at least 10 branches with signals of adaptation. We also selected proteins with i) at least one available tertiary structure, ii) amino acid level resolution of the interaction with one or more viruses, and iii) host tropism.

The most positively selected non-antiviral VIP that fulfills all these requirements is aminopeptidase N, abbreviated ANPEP, APN or CD13 (*Mina-Osorio, 2008*). The analysis of a phylogenetic tree including 84 mammals (*Supplementary file 1J*) confirms pervasive adaptation of ANPEP across mammals, with 76 out of 165 branches in the tree showing signals of adaptation (*Figure 7A*). Note that adaptation of ANPEP has previously been detected in the context of oxidative stress in Cetaceans (*Yim et al., 2014*). ANPEP is a cell-surface enzyme well known for its surprisingly wide range and diversity of functions (*Mina-Osorio, 2008*). In particular, it is used by group I coronaviruses as a receptor, including the Human Coronavirus 229E (HCoV-229E) (*Yeager et al., 1992*), Transmissible Gastroenteritis Virus (TGEV) (*Delmas et al., 1992*), Feline Coronavirus (FCoV) (*Tresnan and Holmes, 1998*), Canine Coronavirus (CCoV) (*Tusell et al., 2007*), Porcine Respiratory Coronavirus (PRCV) (*Delmas et al., 1993*) and Porcine Epidemic Diarrhea Virus (PEDV) (*Oh et al., 2003*). Reguera *et al.* (*Reguera et al., 2012*) solved the structure of porcine ANPEP bound together with TGEV and PRCV.

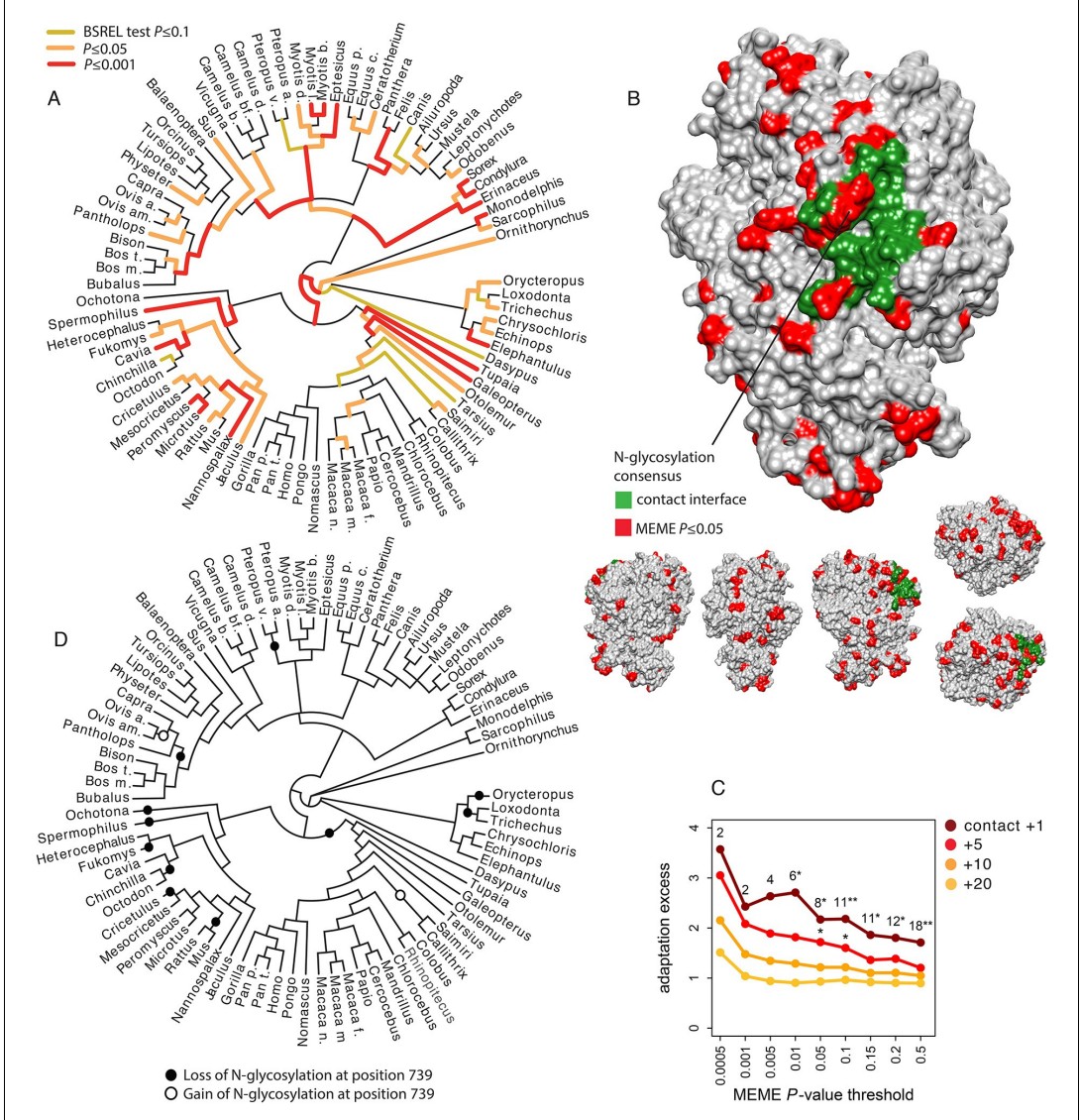

**Figure 7.** Patterns of adaptation to coronaviruses in aminopeptidase N. (**A**) BS-REL test results for ANPEP in a tree of 84 mammalian species. Legend is on the figure. (**B**) Contact surface with PRCV and TGEV on ANPEP structure (PDB 4FYQ). The figure includes visualizations of all the six different faces of ANPEP. Legend is in the figure. (**C**) Excess of adaptation in and near the contact interface with PRCV and TGEV. Within the contact interface plus a given number of neighboring amino acids (one, five, ten or 20 in the figure), adaptation excess (y axis) is defined as the number of observed codons with a MEME *P*-value lower than the *P*-value threshold on the x axis, divided by the average number of codons under the same *P*-value threshold obtained after randomizing the location of adaptation signals over the entire ANPEP coding sequence 5000 times. Dark red curve: adaptation excess within the contact interface with TGEV and PRCV plus one neighboring amino acid. Red curve: plus five neighboring amino acids. Orange: plus ten neighboring amino acids. Light orange: plus 20 neighboring amino acids. Numbers in the figure represent the number of adapting codons, and the stars give the significance of the excess. One star: excess p≤0.05. Two stars: p≤0.01. (**D**) Losses and gains of the N-glycosylation across the mammalian phylogeny.

The authors identified in the extracellular domain of ANPEP 22 amino acids that form a surface of contact with TGEV and PRCV (*Figure 7B*) (*Pettersen et al., 2004*). The most important component of this contact surface for host tropism is a N-glycosylation site at position 736 in porcine ANPEP (orthologous position 739 in human ANPEP) that forms hydrogen bonds with TGEV and PRCV (*Tusell et al., 2007*; *Reguera et al., 2012*). Deleting this site abolishes the ability of TGEV and PRCV to bind porcine ANPEP (*Reguera et al., 2012*). Adding the glycosylation site in human ANPEP that natively lacks it transforms it into a receptor for TGEV and PRCV (*Reguera et al., 2012*).

We use the MEME test from HYPHY (*Murrell et al., 2012*) to identify codons in ANPEP that were under episodic adaptive evolution in mammals. MEME detects significant adaptation (MEME $p \leq 0.05$) in 85 of the 931 aligned codons. Interestingly, several of these adaptively evolving codons are within, or right next to the surface of contact with TGEV and PRCV (*Figure 7B and C*). The codons in contact with TGEV and PRCV and their neighbors are strongly enriched in adaptation compared to ANPEP codons as a whole (*Figure 7C*). This enrichment fades very rapidly as one gets further from the surface of contact with TGEV and PRCV, consistent with detected adaptation being related to interaction with coronaviruses, and not to a more diffuse, less specific enrichment within a wider segment of ANPEP (*Figure 7C*).

Adaptively evolving codons in the contact surface with TGEV and PRCV most notably include two codons within the consensus motif for the N-glycosylation site responsible for host tropism (*Figure 7B*). N-glycosylation is governed by a three amino acids consensus, NX[ST], where X can be anything except proline (*Bause, 1983*). The first and third positions in the consensus evolved adaptively in mammals (MEME *P*=0.005 for both). The ancestral states of the two positions shows that the mammalian ancestor had a fully functional consensus for N-glycosylation, and that the consensus was lost independently 11 times in mammals, either by modification of the first or third position (*Figure 7D*). The consensus was regained only two times after loss (*Figure 7D*). This suggests that the signals of adaptation detected at the first and third positions in the consensus mainly reflect parallel, adaptive losses of the N-glycosylation site in multiple mammalian lineages. Given the crucial role of this N-glycosylation site in the binding of TGEV, PRCV, FCoV and CCoV to ANPEP, it is probable that these parallel adaptive losses were due to selective pressure exerted by ancient coronaviruses.

## Discussion

Here, we have shown that viruses have been a major selective pressure in the evolution of the mammalian proteome. Indeed viruses appear to drive ~30% of all adaptive amino acid changes in the conserved part of the human proteome, as evidenced by the MK test. Furthermore, the footprints of the arms race with viruses are visible in a large number of VIPs, and in a broad range of mammals. Importantly, we find a substantial enrichment in strong signals of adaptation in VIPs with no known antiviral or other immune functions (*Figures 4B* and *5A*). Instead adaptation to viruses is visible in VIPs with a very diverse range of functions including such core functions as transcription or signal transduction (*Figures 4B*, *5C* and *Supplementary file 1C*). This very diverse range of functions strongly argues in favor of a pervasive, external selective pressure –in our case viruses– as the cause for the observed signals.

Our results thus draw a broader picture where adaptation against viruses involves not only the specialized antiviral response, but also the entire population of host proteins that come into contact with viruses. The best-known case of a housekeeping protein having adapted in response to viruses, the transferrin receptor, may thus represent the rule more than an exception (*Demogines et al., 2013*). In line with this, a new non-antiviral protein, NPC1, has very recently been shown to adapt against its use as a receptor by filoviruses in bats (*Ng et al., 2015*).

Although we find a strong signal of increased adaptation, the amount of adaptive evolution that can be attributed to viruses is probably underestimated by our analysis. First, there likely to be many undiscovered VIPs. There is no sign that the pace of discovery of new VIPs is slowing down (*Figure 2*). This means that a substantial number of proteins classified as non-VIPs in this analysis are in fact VIPs, making non-VIPs a conservative control. Second, adaptation in response to viruses is most likely not restricted to proteins that physically interact with viruses. For example, adaptation to viruses might happen in proteins that act downstream of VIPs in signaling cascades, or in non-coding sequences that regulate the expression of VIPs. Third, not all of the 1256 VIPs we use here have been consistently interacting with viruses during evolution. Most VIPs in the dataset (95%) were discovered in humans, and how frequently these VIPs have also been interacting with viruses in other mammals is currently unknown. Some VIPs like PKR have probably been in very frequent contact with diverse viruses. Conversely, other VIPs may have been in contact with viruses for a very limited evolutionary time in mammals, and only in a limited range of lineages. This would apply to VIPs that interact with viruses with a limited host range and few other phylogenetically closely related viruses, as is the case with HCV (*Figure 5B*). In addition, we could only work with VIPs active in current

human populations reflecting the set of viruses infecting humans at present. This means that a potentially large number of proteins classified as non-VIPs in our study were actually VIPs during past human evolution or during the evolution of other mammalian lineages. Altogether, the bias of our sample towards present human VIPs thus makes our results conservative.

We speculate that our results might explain a puzzling observation that rates of protein adaptation appear relatively invariant across different biological functions assessed using GO analysis (*Bierne and Eyre-Walker, 2004*). Because viruses (and likely other pathogens) interact with diverse proteins across most GO functions, they elevate the rate of adaptation across the whole proteome in a way that appears independent of specific functions in the GO analysis. We argue that grouping genes together based on the way they interact with diverse pathogens or other environmental stimuli might be a profitable way for discerning the nature of selective pressures that have molded animal genomes.

In conclusion, our analysis suggests that viruses have exerted a very powerful selective pressure across the breadth of the mammalian proteome, and suggests the possibility that pathogens in general are the key driver of protein adaptation in mammals and likely other lineages and might have driven many pleiotropic effects on diverse biological functions.

## Materials and Mmethods

### Manual curation of virus-interacting proteins

We identified 1256 proteins that physically interact with viruses out of a total of 9,861 proteins with orthologs in the genomes of the 24 mammals included in the analysis (*Figure 1* and *Supplementary files 1A and 1B*). Annotation of the 1256 interacting proteins was performed by querying PUBMED (http://www.ncbi.nlm.nih.gov/pubmed). We started by downloading all the known human gene identifiers gathered by the HUGO Gene Nomenclature Committee (http://www.genenames.org/) (*Gray et al., 2015*) for each of the 9861 orthologs (on the 24th of June 2013). Each gene identifier was then used to automatically query PUBMED together with the term virus. The first automatic query was performed on the 24th of June 2013, and the last update was performed by automatically querying PUBMED again on the 10th of December 2014. We then manually went through all the matches by first looking at the titles of the publications retrieved. Titles of publications unlikely to report an interaction, such as publications about antiviral medications or publications about the prevalence of a virus in a given population, were discarded from the annotation process. Titles of publications directly mentioning an interaction, or of publications reporting insights at the cellular/molecular biology level were retained for further inspection of their abstracts. For the vast majority of cases, interactions between a host protein and a viral protein, RNA or DNA are clearly reported in the abstracts, as it usually represents an important finding of many publications in the virology literature. For more ambiguous cases, we went through the full text of publications. We did not consider interactions identified only through high throughput methods such as two-hybrid or mass spectrometry screens to limit the number of false positives in our dataset (*Mellacheruvu et al., 2013*). In total, we could identify 982 proteins with at least one known interaction with a viral protein, RNA or DNA. We completed our own annotations with 274 additional proteins identified with low throughput methods listed in the VirHostNet (http://virhostnet.prabi.fr/) (*Guirimand et al., 2015*) and HPIDB (http://www.agbase.msstate.edu/hpi/main.html) (*Kumar and Nanduri, 2010*) databases as of February 4th 2015. In total, 1256 of the 9861 orthologous proteins (13%) were found to interact with viral proteins, RNA or DNA according to low throughput methods. These interactions are available online as a supplemental table (*Supplementary file 1A*).

### Multiple alignments of mammalian orthologs

We identified and aligned orthologous coding sequences in the genomes of 24 mammals (*Figure 1* and *Supplementary file 1B*). Those 24 mammals were those with high sequencing depth genomes as of December 2012. They include the assemblies hg19 for human, chimpanzee panTro4, gorilla gorGor3, orangutan ponAbe2, gibbon nomLeu3, macaque rheMac3, baboon papAnu2, marmoset calJac3, bushbaby otoGar3, mouse mm10, rat rn5, guinea pig cavPor3, squirrel speTri2, rabbit oryCun2, sheep oviAri3, cow bosTau7, pig susScr3, microbat myoLuc2, panda ailMel1, ferret musFur1, dog canFam3, cat felCat5, horse equCab2, and elephant loxAfr3. We first used Blat (*Kent, 2002*) to

find all the homologous matches of 22,074 human coding sequences (CDS) from Ensembl v69 (*Flicek et al., 2012*) in all the assemblies listed above. These CDS are the longest CDS for their respective human genes. Blat was parameterized for high sensitivity, using translated genomes (options -q=dnax and -t=dnax), translated queries and setting the minimum identity to 50%, in addition to using the -fine option. Best Blat matches with the highest number of matching positions were then blatted back on the human genome to identify best reciprocal hits. This way we could find a total of 9861 human CDS with best reciprocal hits in all the other 23 species, no in-frame stop codon and at least 30% of the length of the human CDS (*Supplementary file 1B*). Of these, 9338 human CDS were found to have clear conserved synteny in at least 18 of the 23 non-human species, with at least five conserved neighboring protein coding genes within 500 kb upstream their 5'start and 500 kb downstream their 3' end (*Supplementary file 1B*). The CDS of these 9338 orthologs were then aligned using PRANK under the codon evolution model (*Loytynoja and Goldman, 2008*). Any codon present in less than eight species was discarded. The combination of Blat to extract the homologous sequences and PRANK to realign them ensures we work with high quality alignments. Indeed, the first step of local alignment with Blat excludes segments of the CDS that are too diverged between mammals to be properly aligned during the subsequent step of global alignment with PRANK, which has been shown to be the best coding sequence aligner available (*Fletcher and Yang, 2010*; *Jordan and Goldman, 2012*).

## Annotation of antiviral and immune mammalian orthologs

We consider proteins as antiviral if they restrict viral replication in any way, for example by directly engaging viruses for recognition and/or degradation of viral molecules, or if they have been shown to be required for the proper unfolding of the antiviral response. Antiviral proteins were annotated at the same time as VIPs and are listed in *Supplementary file 1D*.

We also annotated VIPs with immune functions while identifying VIPs. The final set of immune VIPs is made of the 203 VIPs annotated with the GO categories 'immune system process' (GO:0002376), 'defense response' (GO:0006952) or 'immune response' (GO:0006955) (2015), in addition to 38 immune VIPs we annotated based on publications reporting roles in various parts of the immune response, from innate to adaptive immune response, from regulators to effectors of the immune response, to VIPs involved in the development of the immune system. *Supplementary file 1D* lists all the immune VIPs identified.

## Quantifying adaptation in human with the classic and the asymptotic McDonald-Kreitman tests

We estimate and compare the proportion of adaptive amino acid changes, noted $\alpha$, in VIPs and non-VIPs using either the classic McDonald-Kreitman test (MK test) (*McDonald and Kreitman, 1991*) or an asymptotic MK test (*Messer and Petrov, 2013*). The MK test measures $\alpha$ as follows:

$$\alpha = 1 - (\mathrm{DS} * \mathrm{pN})/(\mathrm{DN} * \mathrm{pS})$$

where DN and DS are the number of fixed amino acid substitutions and synonymous substitutions in the lineage studied since divergence with a closely related species, and pN and pS are the number of polymorphic non-synonymous and synonymous sites, respectively. For our study, DN and DS are the number of substitutions in the human lineage since divergence with chimpanzee. Theses substitutions are identified in human-chimpanzee-orangutan alignments of the mammalian orthologous CDS (*Supplementary file 1E*). These alignments are prepared the same way as the alignments of 24 mammals (see Materials and methods, 'Multiple alignments of mammalian orthologs'), except for the fact that we include all isoforms of every ortholog in the analysis. The substitutions are those positions where chimpanzee and orangutan have the same nucleotide but human has a different nucleotide. We further require the substitution to be fixed in human, as shown by the fact that the position is not polymorphic in the 1000 Genomes Project data (*Abecasis et al., 2012*). The measures of pN and pS are obtained as the numbers of polymorphic positions in African populations from the final phase I 1000 Genomes Project. For the classic MK test, we only use variants with a derived allele frequency between 0.1 and 0.9 in African populations in order to limit the effect of deleterious mutations. Note however that the classic MK test has been shown to underestimate $\alpha$ even when excluding low frequency variants. The classic MK test is thus well suited to measure the relative

difference in adaptation between VIPs and non-VIPs, but it is not well suited to actually measure the true absolute $\alpha$ in VIPs and non-VIPs.

For this we use the asymptotic MK test since it does not underestimate $\alpha$ (*Messer and Petrov, 2013*). It is nevertheless limited to using large numbers of genes (>1000). The asymptotic MK test is robust to the presence of deleterious mutations and to demography. The asymptotic MK test works by estimating $\alpha$ in bins of derived allele frequencies. For example, $\alpha$ can be calculated in the bin of frequencies from 0.1 to 0.2 by counting only variants with a derived allele frequency between 0.1 and 0.2 to measure pN and pS. An exponential curve is then fitted to the estimates of $\alpha$ across bins of frequency. The value taken by the fitted curve for a derived allele frequency of 100% provides the estimate for $\alpha$ (*Messer and Petrov, 2013*). Using the asymptotic MK test, Messer and Petrov (*Messer and Petrov, 2013*) estimated that $\alpha$ is 57% in *Drosophila melanogaster*, and 13% in human. For both species, these estimates were obtained based on polymorphism and divergence data for most of the proteome (more than $10^4$ protein coding sequences in both cases). Here, we need to estimate $\alpha$ in 1256 VIPs, and in the same number of randomly sampled non-VIPs. This is an order of magnitude less than the number of coding sequences used by Messer and Petrov (*Messer and Petrov, 2013*), which makes curve fitting challenging. Indeed, the low number of high frequency variants means that estimates of $\alpha$ in the high frequency bins are very noisy. Using the stable release 1000 Genomes Project final phase I variants from African populations, we count that VIPs only have 143 non-synonymous and 343 synonymous variants with a derived allele frequency above 0.5, respectively. The [0.8,0.9] bin of frequency only has 18 non-synonymous and 51 synonymous variants. In comparison, the [0.1,0.2] bin has 149 non-synonymous and 370 synonymous variants. The [0.4,0.5] bin still has twice more variants (36 non-synonymous, 101 synonymous) than the [0.8,0.9] bin.

The low number of high frequency variants is however not the only issue. A second potential issue when trying to fit a curve to predict $\alpha$ in the asymptotic McDonald-Kreitman test is the mispolarization of alleles as ancestral or derived. Mispolarization is a common problem that distorts the unfolded Site Frequency Spectrum (SFS) (*Hernandez et al., 2007*). The most severe distortion is usually within the high frequency part of the SFS (*Hernandez et al., 2007*). Indeed, abundant low-frequency derived variants are often misidentified as high frequency derived variants. This can result in substantial overestimations of the number of high frequency variants. The number of non-synonymous variants pN might be more severely overestimated than pS, since less high frequency and more low frequency non-synonymous variants are expected in the first place. This could hypothetically result in underestimates of $\alpha$ within high frequency bins.

Here we modify the asymptotic MK test to circumvent the mispolarization and high frequency, high noise issues. We do so by estimating $\alpha$ based only on derived allele frequencies lower than 0.5, where the distortion of the SFS due to mispolarized alleles is negligible. This also makes the asymptotic McDonald-Kreitman less reliant on bins of high frequencies with very noisy estimates of $\alpha$, due to small values of pN and pS.

We use either a logarithmic fit of the form $y = a + b(\ln(x + c))$ over the range of frequencies 0 to 0.5 (*Figure 4C*), or an exponential fit of the form $y = a + b * \exp(-x/c)$. Both the logarithmic fit and the exponential fits provide accurate estimates of $\alpha$ for a wide range of evolutionary scenarios, as shown by forward population simulations using SLIM (*Messer, 2013*).

We use the forward population simulator SLIM (*Messer, 2013*) to simulate a typical, 400 codons, six exons coding sequence. Each exon is separated by 4000 bp long introns. One in four coding sites is synonymous and only experiences neutral mutations. Non-synonymous sites experience neutral, advantageous, strongly deleterious and slightly deleterious mutations. The coding sequence evolves for 20,000 generations in a population of 1000 individuals, at a uniform mutation rate of $2.5 \times 10^{-7}$ and with a uniform recombination rate of 10 cM/Mb. These parameters are equivalent to 200,000 generations of evolution of a 10,000 individuals population with a mutation rate of $2.5 \times 10^{-8}$ and a recombination rate of 1 cM/Mb. This results roughly in the amount of divergence observed in the human lineage since divergence with chimpanzee. The rescaling by a factor of ten greatly speeds up the simulations. Roughly matching the observed DN, DS, pN and pS in VIPs requires simulating 1000 coding sequences. The true $\alpha$ obtained from simply counting adaptive fixations in the simulations can then be compared with the $\alpha$ estimated from DN, DS, pN and pS. By repeating the simulation of sets of 1000 coding sequences many times, we can get the variance of the estimation of $\alpha$ both by the modified asymptotic MK test. By repeating the simulations 100 times, we show that the modified asymptotic MK test gives accurate estimates of $\alpha$ for all evolutionary scenarios tested

(*Supplementary file 1G*). In practice, the logarithmic fit is easier to use than the exponential fit. Indeed, fitting algorithms such as the ones implemented in the LM() function or the nlsLM() function from the minpack.lm package in R often fail to converge for the exponential fit. We therefore use the logarithmic fit.

## Quantifying adaptation in the mammalian phylogeny

We use the multiple alignments of the coding sequences from the 24 mammals listed above to quantify adaptation across mammals. There are three different types of tests aimed at detecting and quantifying adaptation in a multi-species coding sequence alignment: branch tests, site tests, and branch–site tests.

The so-called branch tests look for branches in a tree where the ratio of non-synonymous to synonymous substitutions dN/dS exceeds one for the entire coding sequence. In order to happen this requires an extreme amount of adaptation in a specific branch. Branch tests thus detect only the most extreme bursts of adaptation, and have very low statistical power to detect the vast majority of more moderate bursts of adaptation in a phylogeny (*Nielsen et al., 2005*). This makes them a very poor choice to quantify adaptation within an entire phylogeny.

Site tests look for specific codons of a coding sequence where dN/dS significantly exceeds one across the entire phylogeny. Codons with dN/dS >> 1 are codons that have accumulated many adaptive non-synonymous substitutions across the tested phylogeny. This means site tests ignore the case where specific codons have evolved adaptively on a specific branch, probably the most common case in coding sequence evolution (*Murrell et al., 2012*). Although site tests are well suited for cases where there is a strong *a priori* expectation about which sites should evolve adaptively, as is for example the case of TFRC, here we have no a priori knowledge about the sites that are expected to evolve adaptively in VIPs in response to viruses. Instead we use branch-site tests which are designed to detect adaptation at specific codons in specific branches.

There are currently two main implementations of the branch-site test, one available in PAML (*Zhang et al., 2005*, *Yang, 2007*) and one available in the HYPHY package (*Kosakovsky Pond et al., 2011*). The two tests are both likelihood ratio tests that compare a model integrating positive selection with a neutral model without positive selection. The PAML branch-site test and the HYPHY BS-REL branch-site test differ mainly in the assumptions of their evolutionary models. The PAML branch-site test defines two kinds of branches in the phylogenetic tree used, the foreground and background branches. The foreground branch is the branch where the presence of positive selection is tested. The evolutionary model of the branch-site test authorizes positive selection in the foreground branch, but not in the background branch. Unlike the PAML branch-site test, the HYPHY BS-REL test uses a model that has no limitation regarding the occurrence of adaptation across the tree. This difference in the models used has very profound consequences for the ability of the two tests to detect and quantify recurrent adaptation (*Kosakovsky Pond et al., 2011*). Indeed, the HYPHY test has good power to detect recurrent adaptation. Because it does not allow adaptation in the background branches, the PAML tests suffers a severe loss of statistical power when recurrent adaptation does occur in the background branches. As an example, the HYPHY BS-REL test detects significant (BS-REL test p≤0.05) signals of adaptation in 18 branches of the mammalian tree used in this study for PKR (*Figure 6*). In comparison, the PAML test detects only nine branches (PAML test p≤0.05). This is a crucial difference between the two tests in our case given that the arms race with viruses is likely to trigger recurrent bursts of adaptation across mammals. For this reason we use HYPHY BS-REL to quantify adaptation in mammals. More specifically, we use the proportion of selected codons estimated by the BS-REL test to quantify adaptation. To estimate the strength of the evidence in favor of adaptation across the entire mammalian tree, we use the *P*-value of the BUSTED test in HYPHY that uses the same codon evolution model as the BS-REL test.

In this study, we compare VIPs and non-VIPs for weak (BUSTED *P*-values ≤0.9) to increasingly strong (BUSTED *P*-values ≤$10^{-5}$) evidence of adaptation. We start by computing the average proportion of codons under adaptive evolution for VIPs and the same average proportion for the sets of randomly matched non-VIPs (see the description of the permutation test). For each coding sequence, we retrieve the proportion of positively selected codons on each branch, and compute the average of this proportion across branches. More specifically, we only count branches of the tree with conserved synteny (*Supplementary file 1B*). That is, if 40 of the 44 branches of the tree have conserved synteny (see above), we compute the average proportion of selected codons only from

these 40 branches. In practice we tolerate only up to five branches in the tree with no conserved synteny (at least 39 branches with conserved synteny; *Supplementary file 3*). This reduces the dataset of orthologs that can be used in the analysis only slightly, from 9861 to 9338 total, and among those the number of VIPs from 1256 to 1193. Then adaptation is simply quantified as the average proportions of selected codons in valid branches across VIPs (or the same number of matched non-VIPs). If the threshold for BUSTED *P*-value is set to $10^{-x}$, we only include in the quantification the average proportions of selected codons from coding sequences with BUSTED *P*-value$\leq 10^{-x}$. For a low x, we compare how much selection occurred in VIPs and non-VIPs counting both weak and strong signals of adaptation. For a high x, we compare how much strong, highly significant signals of adaptation occurred in VIPs compared to non-VIPs.

## The pN/pS ratio as a measure of purifying selection

We designed a permutation test that makes it possible to compare adaptation in VIPs with adaptation in non-VIPs with the same amount of purifying selection. The amount of purifying selection in a protein corresponds to the proportion of amino acids that cannot change, or very infrequently during evolution. On average VIPs experience much more purifying selection than non-VIPs. This means that mechanically, a smaller proportion of amino acids can possibly be targeted by adaptive evolution in VIPs. A naïve comparison of VIPs and non-VIPs would therefore tell more about the difference in purifying selection than about the difference in the amount of adaptation. Instead, the idea is to compare adaptation in the 1256 VIPs with adaptation in 1256 non-VIPs with the same overall average and variance in levels of purifying selection. The sampling of non-VIPs with similar purifying selection VIPs is however challenging.

The first question is which measure of purifying selection to use? The ratio of non-synonymous to synonymous substitution rates dN/dS is often used as a measure of purifying selection. How much smaller dN is compared to dS can indeed tell how evolutionarily constrained a protein is. However the problem with dN/dS is that dN not only reflects purifying selection, but also reflects adaptive amino acid substitutions. This means that comparing VIPs with non-VIPs with similar dN/dS ratios would underestimate an excess of adaptation in VIPs. This is because more adaptation would increase dN/dS in VIPs more than it does in non-VIPs (*Figure 5—figure supplement 1*; compare VIP with non-VIP I). As a result, VIPs would be matched with either non-VIPs that have the same dN/dS because they have the same levels of both purifying selection and adaptation (*Figure 5—figure supplement 1*; compare VIP with non-VIP II), or non-VIPs that have the same dN/dS because they have experienced less purifying selection, and thus have also had more codons available for adaptation to happen (*Figure 5—figure supplement 1*; compare VIP with non-VIP III), or non-VIPs that have the same dN/dS because they are under both stronger purifying selection and adaptation (*Figure 5—figure supplement 1*; compare VIP with non-VIP IV). In all three cases, non-VIPs selected as controls would experience more adaptation than non-VIPs that would have been matched based on a more accurate measure of purifying selection (*Figure 5—figure supplement 1*; compare non-VIP I with non-VIPs II, III and IV). This ultimately results in underestimating any excess of adaptation in VIPs compared to non-VIPs.

Unlike the dN/dS ratio, the ratio of non-synonymous to synonymous polymorphism pN/pS only reflects purifying selection. Indeed, pN is decreased by purifying selection, but is not affected by adaptive mutations that segregate for very short times in populations. This makes pN/pS a much better measure of purifying selection than dN/dS that can be used to match VIPs with similarly constrained non-VIPs. There are however two problems with the pN/pS ratio.

The first is that proteome-wide estimates of pN/pS are not available for all the mammals included in this analysis. Good estimates of pN/pS require sequencing the genomes of a sufficient number of non-inbred individuals, ideally more than ten, within a given species. The pN/pS ratio is publicly available, based on the genome sequences of a sufficient number of individuals, in human (*Abecasis et al., 2012*) and the non-human primate species represented in the Great Ape Genome Project, namely chimpanzee, gorilla and orangutan (*Prado-Martinez et al., 2013*). The limited number of species of the mammalian tree with pN/pS information can still be used as a control of purifying selection in the permutation test for all mammals. It is true that the pN/pS ratio within a species or a subset of species does not represent the absolute, overall level of purifying selection in the entire mammalian tree. It is known for instance that primates experience weaker purifying selection than rodents. What matters however for the permutation test is not the absolute level of purifying

selection, but the relative difference in pN/pS between VIPs and non-VIPs. Indeed, VIPs and matched non-VIPs with similar pN/pS experience similar purifying selection, even if pN/pS is from a subset of species in the mammalian tree. Whether pN/pS is overall skewed towards higher or towards lower values in the subset of species used, then the skew is still the same for both VIPs and non-VIPs. This means that the relative difference in pN/pS is still a good measure of the general difference in purifying selection across mammals. A different skew in VIPs and non-VIPs requires invoking unlikely scenarios where VIPs would experience a global relaxation or intensification of constraint specifically in the primate species where pN/pS is available. Given the high number of VIPs and their high functional diversity (*Supplementary file 1C*), such a global trend towards relaxation or higher constraint in primates is extremely unlikely. The pN/pS ratio from primate species can therefore be used as a control for purifying selection.

More specifically, we use the pN/pS ratio from populations of chimpanzees (Nigeria-Cameroon, Eastern and Central populations), gorillas (Western lowland population) and orangutans (Sumatran and Bornean populations) (*Supplementary file 1F*). These populations are the populations included in the Great Apes Genome Project with the highest effective population sizes. Indeed, the pN/pS ratio is less noisy and available for more proteins in populations with higher population sizes and higher genetic diversity. For each VIP and non-VIP, the value of pN/pS used in the permutation test is simply the sum of pN across all the primate populations divided by the sum of pS across the same populations (*Supplementary file 1F*). In each primate population, pN and pS are measured excluding singletons to limit the influence of potential erroneous variant calls.

The second potential problem with pN/pS is that it is a noisy measure of purifying selection. At any time in a population of primates, only few positions are polymorphic within a typical (~300 codons) coding sequence. As a consequence, a highly constrained coding sequence may by chance have more non-synonymous variants than synonymous variants, and a high pN/pS ratio. Conversely, a weakly constrained coding sequence may by chance have less non-synonymous variants, and a low pN/pS ratio. This is problematic if we want to use pN/pS as a control for purifying selection. One can consider the case of VIPs where pN/pS is substantially lower than in non-VIPs. Matching VIPs with non-VIPs with a similarly lower pN/pS, we would end up selecting non-VIPs with a lower pN/pS not because of purifying selection but merely because of noise. This makes controlling for purifying selection less straightforward than directly matching each individual VIP with non-VIPs with similar pN/pS ratios. Instead we use an indirect matching strategy.

## Permutations with a target average: the example of purifying selection

As described above, the pN/pS ratio is a noisy measure of purifying selection. This means that we cannot use a direct matching strategy between VIPs and non-VIPs for the permutation test. An indirect matching strategy can however still be used, that uses the mammals-wide rate of non-synonymous substitutions dN as an intermediate. In particular, we use PAML to estimate dN and dS under the M8 evolution model (*Yang, 2007*). The dN/dS ratio for the whole mammalian tree (*Supplementary file 1B*) integrates hundreds of millions of years of evolution. In the absence of adaptation, it would therefore be a much less noisy measure of purifying selection than pN/pS. The issue is however that dN is influenced by both purifying selection and by adaptation. If VIPs experience more adaptation than non-VIPs, then purifying selection being equal, we expect dN/dS to be higher in VIPs than in non-VIPs. If VIPs experience less adaptation than non-VIPs, then purifying selection being equal, we expect dN/dS to be lower in VIPs than in non-VIPs.

VIPs have a 25% lower pN/pS than non-VIPs in great apes, but only a 15% lower dN/dS than non-VIPs. The smaller difference in dN/dS than pN/pS is an indication that adaptation has increased dN more strongly in VIPs than in non-VIPs. Purifying selection being equal, dN/dS is therefore higher in VIPs than in non-VIPs. This means that non-VIPs with the same pN/pS (purifying selection) as VIPs have a lower dN/dS. We can therefore match VIPs and non-VIPs with the same pN/pS by selecting non-VIPs with a dN/dS ratio that is only a fraction of the dN/dS in VIPs. This fraction can be adjusted through trial and error until finding the one that matches VIPs with non-VIPs with the same overall average pN/pS. This indirect matching strategy makes it possible to compare VIPs and non-VIPs with the same level of purifying selection while avoiding the pitfall of noise in pN/pS.

The random sets of non-VIPs must fulfill two criteria to be comparable to VIPs. First, non-VIPs should have the same overall average pN/pS as VIPs. Second, non-VIPs should have the same variance in pN/pS, *i.e.* pN/pS values in non-VIPs are spread as much as they are in VIPs. This can all be

achieved by using a permutation scheme where samples of non-VIPs must satisfy a pre-fixed, target average (*Figure 5—figure supplement 2*). Note that although here we detail the case of purifying selection, permutations with a target average can be used to get samples of non-VIPs similar to VIPs for any possible factor.

For the case of purifying selection, the permutations with an average target work as follows. We first measure the average dN in VIPs, noted $dN_{(vip)}$. Then we define the target average dN we wish the chosen non-VIPs to exhibit at the end of the sampling. In specific, we ask the target dN to be a fraction $a$ of $dN_{(vip)}$, plus or minus 5%. The target dN for non-VIPs can therefore take values between $dN_{(inf)}$ and $dN_{(sup)}$, where $dN_{(inf)} = 0.95(adN_{(vip)})$ and $dN_{(sup)} = 1.05(adN_{(vip)})$. The fraction $a$ is set manually through trial and error so that the sampled non-VIPs have the same average pN/pS as VIPs. Note that we use dN and not dN/dS to avoid giving too much weight to dS, as it tends to saturate and take much greater values than dN (*Supplementary file 1B*) and thus bears much more heavily on the dN/dS ratio.

Non-VIPs are sampled using a simple algorithm described in *Figure 5—figure supplement 2*. We first randomly sample a set of five non-VIPs. This initial sampling of five non-VIPs is repeated until their average dN falls within the target interval $[dN_{inf}, dN_{sup}]$. We then add randomly sampled non-VIPs one at a time until their number matches the number of VIPs. The average of all the sampled non-VIPs has to remain within $[dN_{(inf)}, dN_{(sup)}]$ (blue dots in *Figure 5—figure supplement 2*), except for every X non-VIP that is sampled completely randomly (red dots in *Figure 5—figure supplement 2*). This means that in the latter case the average dN of the sampled non-VIPs can fall out of $[dN_{(inf)}, dN_{(sup)}]$. When this happens we sample non-VIPs with dN values that bring the average dN back within $[dN_{(inf)}, dN_{(sup)}]$ (grey dots in *Figure 5—figure supplement 2*). That is, if the average dN is above $dN_{(sup)}$, we sample as many non-VIPs as necessary that each lower the average dN until it falls back within $[dN_{(inf)}, dN_{(sup)}]$. If the average dN is below dN(inf), we sample non-VIPs that each increase the average dN until it falls back within $[dN_{(inf)}, dN_{(sup)}]$. The parameter X is the parameter of the test that makes it possible to match the variance in pN/pS of the sampled non-VIPs with the variance observed for VIPs. A low X gives samples of non-VIPs with a higher variance. A high X gives samples of non-VIPs with a lower variance.

To define the fraction $a$ and X, we get $10^4$ random samples of non-VIPs. We then test whether the average and variance of pN/pS in VIPs are significantly different or not from the distributions of averages and variances of pN/pS given by the $10^4$ random samples of non-VIPs. We find that $a=0.7$ and X=3 give samples of non-VIPs with slightly significantly higher average pN/pS than VIPs' pN/pS (0.56 in non-VIPs versus 0.526 in VIPs, *P*=0.03) and a very similar variance (0.53 in non-VIPs vs 0.552 in VIPs, *P*=0.35). The pN/pS ratio in human african populations is also slightly higher in non-VIPs compared to non-VIPs (0.81 in non-VIPs compared to 0.76 in VIPs, *P*=0.04), which shows that our calibration is robust to the species used to measure pN/pS. There is no combination of $a$ and X where both the average and variance of pN/pS are identical in VIPs and the sampled non-VIPs. The combination of $a=0.7$ and X=3 gives the closest matching variances and a slightly higher pN/pS (lower purifying selection, see above for numbers) in non-VIPs than in VIPs. Other combinations give closer averages of pN/pS, but more distant variances. To be conservative, we thus choose to use $a=0.7$ and X=3. The fact that the sampled non-VIPs experience slightly less purifying selection than VIPs makes the comparison conservative (the less purifying selection in non-VIPs compared to non-VIPs, the more opportunities there were for adaptation to happen at positions of coding sequences that can change). Finally, using $a=0.97$ and X=2, we can compare VIPs and non-VIPs with similar dN/dS ratios (VIPs' and non-VIPs' average dN/dS=0.124, *P*=0.51). As expected (*Figure 5—figure supplement 1*), using matching dN/dS instead of pN/pS strongly underestimates, but yet still reveals a substantial excess of adaptation in VIPs compared to non-VIPs (39% adaptation excess, *P*=0 versus 117% excess, *P*=0 after $10^9$ iterations, when matching pN/pS; see *Figure 5A*).

## Gene Ontology-matching control samples

Throughout this analysis we distinguish between the effects due to viruses and the effects due to the functional roles that VIPs play in the host. This is done by comparing VIPs with matching control sets of non-VIPs with similar Gene Ontology (GO) processes. There are 162 GO processes with 50 or more VIPs (*Supplementary file 1C*). The matching procedure is conducted using only these 162 processes. For each VIP, we find all the non-VIPs that have at least 60% of GO processes in common, and where the total number of processes does not exceed 140% of the number in the VIP to be

matched with. We then randomly choose one non-VIPs among all those that fulfill these requirements. With the parameters used, we find each VIP always has more than 5 non-VIPs to choose from, and many more for most VIPs. Furthermore, these parameters give control sets of non-VIPs with representations of GO processes very similar to their representation in VIPs. On average the representation of each GO process is only 18% lower or higher in the matching controls, versus 60% lower or higher in non-matching, randomly sampled sets of non-VIPs. Note that perfect matching is impossible to achieve given that different proteins can have very different and specific combinations of associated GO processes.

## Acknowledgements

We thank Kerry Samerotte, Sandeep Venkataram, Emily Ebel, Pleuni Pennings, Hunter Fraser, Sara Sawyer and Sergei Kosakovsky Pond for comments on the manuscript. This work is funded by NIH grants R01GM089926 and R01GM097415 to DAP.

## Additional information

### Funding

| Funder | Grant reference number | Author |
|---|---|---|
| National Institutes of Health | R01GM089926 | Dmitri A Petrov |
| National Institutes of Health | R01GM097415 | Dmitri A Petrov |

The funders had no role in study design, data collection and interpretation, or the decision to submit the work for publication.

### Author contributions

DE, Conceived and designed the experiments, Performed the experiments, Interpreted the results, Wrote the paper, Acquisition of data; LC, CG, Performed the experiments, Acquisition of data, Analysis and interpretation of data; DAP, Conceived and designed the experiments, Interpreted the results, Wrote the paper

### Author ORCIDs

David Enard, http://orcid.org/0000-0003-2634-8016

## Additional files

### Supplementary files

• Supplementary file 1. (A) Table with VIPs. Interactions are described in the third column. For example, 19386720-EBV-dsDNA means that the article with PUBMED ID 19386720 describes an interaction between a mammalian host protein and an Epstein-Barr Virus EBV protein. The dsDNA label is for the fact that EBV is a double-stranded DNA virus (we use ssRNA for single–stranded RNA viruses, ssRNART for single-stranded RNA retroviruses, dsDNA for double-stranded viruses, dsDNART for double-stranded DNA retroviruses and ssDNA for single-stranded DNA viruses). If the interaction in the example was with EBV's RNA, we would have 19386720-rna-EBV-dsDNA instead of 19386720-EBV-dsDNA. If the interaction was with EBV's DNA, we would have 19386720-dna-EBV-dsDNA. (B) Table with the mammalian orthologous CDS information. The table contains the synteny information as well as the mammals-wide rates dN and dS for each of the 9,861 orthologs included in the analysis. (C) Table with GO categories with more than 50 VIPs. The table includes the information about the 162 GO biological processes with 50 or more VIPs. (D) Table with the 241 VIPs annotated as immune based on GO annotations. (E) Table with the human polymorphism and divergence information for the McDonald-Kreitman test. (F) Table with pN and pS in great apes. (G) Table with results of the forward simulations for the asymptotic MK test. The brackets give the 95% confidence intervals for the predicted α. Advantageous s is the selection coefficient of advantageous mutations. Deleterious s is the selection coefficient of deleterious mutations. (H) Excess of adaptation in human in highly expressed VIPs and non-VIPs at the RNA or protein levels, VIPs and non-VIPs with a high

number of protein-protein interactions. (I) Table with mammalian orthologs and evidence of adaptation. The table provides all the 9,861 orthologs with best reciprocal hits (Material and methods), The first column is Ensembl Gene ID, the second column is BUSTED *P*-value, and the other columns are BS-REL estimated proportions of selected codons in all the 44 branches tested. Note that the proportions of selected codons are set to zero for those branches where there is no good synteny information (Materials and methods). (J) Table with Genbank identifiers for the 84 mammals ANPEP alignment.

### Major datasets

The following dataset was generated:

| Author(s) | Year | Dataset title | Dataset URL | Database, license, and accessibility information |
|---|---|---|---|---|
| Enard D, Cai L, Gwennap C, Petrov DA | 2015 | Data from: Viruses are a dominant driver of protein adaptation in mammals | http://dx.doi.org/10.5061/dryad.fs756 | Available at Dryad Digital Repository under a CC0 Public Domain Dedication |

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
