## [Decision Letter]

Thank you for submitting your work entitled "Viruses are a dominant driver of protein adaptation in mammals" for consideration by *eLife*. Your article has been favourable evaluated by Detlef Weigel (Senior Editor) and three reviewers, one of whom, Gil McVean, is a member of our Board of Reviewing Editors.

The reviewers have discussed the reviews with one another and the Reviewing Editor has drafted this decision to help you prepare a revised submission.

The following individuals involved in review of your submission have agreed to reveal their identity: Sara Sawyer and Sergei Kosakovsky-Pond (peer reviewers).

Summary:

This paper presents molecular evolution data to support a hypothesis that interactions between viral proteins and human (or more generally mammalian) proteins has driven at least 30% of adaptive substitutions in protein-coding genes. The work is based on an extensive and hand-curated database of viral interacting proteins. Phylogenetic analysis of gene alignments and analysis of nonsynonymous and synonymous polymorphism from humans and great apes are used to support the claims. A specific example of adaptation at ANPEP is provided to give some detail on the nature of the insights.

The claim made in this paper is bold and such analyses are hard to make fully watertight given the array of possible confounders. However, the variety of evidence presented, the care taken to deal with known potential confounders, the specific example shown and the interesting evidence from HCV all go a long way to provide a convincing case.

Essential revisions:

1) It is likely that others will want to pore over the claims made in the paper. It is therefore essential that all the information used to support the hypothesis – gene alignments, dN, dS, pN, pS, and VIP data are made available.

2) The multiplication of p-values from BS-REL is not a statistically valid summary of whether there is any evidence for adaptation across branches in the tree. There are other approaches – such as BUSTED (http://www.ncbi.nlm.nih.gov/pubmed/25701167) – which test this hypothesis formally. It also seems likely that other methods – such as REL/FEL in HyPhy or PAML could provide more detailed indications of likely sites of adaptation.

3) Some work needs to be taken to either simplify the diversity of different tests/estimates presented or explain differences between reported values. For example, several different approaches are used to estimate the excess or fraction of nonsynonymous substitutions driven by viral interaction – it would be useful to have a consensus of these and a discussion of variance among estimators. A related point is that additional information about how the excess adaptation is estimated should be provided.

4) The discussion is poorly focused and makes strong statements without support – it would be better to provide closer discussion of the work presented in the paper rather than speculating about future findings. There is also a confusing or confused statement that "the war against viruses is a global war that involves not only the specialised soldiers of the antiviral response, but also the entire population of host proteins that come into contact with viruses". If proteins adapt in response to contact, it is surely more likely they are evading the viral response – likely at some cost to the original function of the protein.

---

## [Author Response]

*1) It is likely that others will want to pore over the claims made in the paper. It is therefore essential that all the information used to support the hypothesis – gene alignments, dN, dS, pN, pS, and VIP data are made available.*

We are sorry that there was an issue for the reviewers to access the data we had made available with our submission. Our submission is now accompanied by one supplementary file. In addition, the alignments of mammalian coding sequences have been made available for the reviewers on Dryad and are referenced in the manuscript:

http://datadryad.org/review?doi=doi:10.5061/dryad.fs756,doi:10.5061/dryad.fs756.

*2) The multiplication of p-values from BS-REL is not a statistically valid summary of whether there is any evidence for adaptation across branches in the tree. There are other approaches – such as BUSTED (http://www.ncbi.nlm.nih.gov/pubmed/25701167) – which test this hypothesis formally. It also seems likely that other methods – such as REL/FEL in HyPhy or PAML could provide more detailed indications of likely sites of adaptation.*

We thank the reviewers for pointing out this inaccuracy in our manuscript. We have now reanalyzed our data with BUSTED, and find that BUSTED supports our previous conclusions and gives even better results than BS-REL (Figure 5, formerly 2). We added the BUSTED results to our manuscript as a new Figure 5. In this figure, we use the BUSTED p-values as a threshold instead of the old product of p-values from BS-REL, but we still use the BS-REL proportions of selected codons because we want to be able to (i) exclude the branches with questionable synteny as explained in the Methods and (ii) we want to be able to look at the excess of adaptation within specific mammalian clades as done in Figure 5.

We did not conduct the site tests analyses suggested by the reviewers for several reasons. First, running the additional BUSTED analysis alone was computationally extremely costly and intensive and explains the rather long time taken for the revision. Second, a comprehensive study of the specific sites under adaptation in VIPs is a full project on its own that we plan to conduct in the future.

*3) Some work needs to be taken to either simplify the diversity of different tests/estimates presented or explain differences between reported values. For example, several different approaches are used to estimate the excess or fraction of nonsynonymous substitutions driven by viral interaction – it would be useful to have a consensus of these and a discussion of variance among estimators. A related point is that additional information about how the excess adaptation is estimated should be provided.*

We would like to apologize for the confusion. We have clarified the reasons why we use different tests and datasets. We wrote a new overview at the beginning of the Results section. We have re-organized the human adaptation part in a clearer way, with subparts corresponding to the different tests we used. The mammalian adaptation part was also broken down into subparts to improve clarity.

*4) The discussion is poorly focused and makes strong statements without support – it would be better to provide closer discussion of the work presented in the paper rather than speculating about future findings. There is also a confusing or confused statement that "the war against viruses is a global war that involves not only the specialised soldiers of the antiviral response, but also the entire population of host proteins that come into contact with viruses". If proteins adapt in response to contact, it is surely more likely they are evading the viral response – likely at some cost to the original function of the protein.*

We have now removed the statement that reviewers found out of focus and too speculative. In addition we removed the entire paragraphs about pleiotropy and about the greater statistical power for paleovirology since they are indeed speculative.